# Causal meets Submodular: Subset Selection with Directed Information

**Yuxun Zhou**
Department of EECS
UC Berekely
yxzhou@berkeley.edu

**Costas J. Spanos**
Department of EECS
UC Berkeley
spanos@berkeley.edu

## Abstract

We study causal subset selection with *Directed Information* as the measure of prediction causality. Two typical tasks, causal sensor placement and covariate selection, are correspondingly formulated into cardinality constrained directed information maximizations. To attack the NP-hard problems, we show that the first problem is submodular while not necessarily monotonic. And the second one is "nearly" submodular. To substantiate the idea of approximate submodularity, we introduce a novel quantity, namely *submodularity index (SmI)*, for general set functions. Moreover, we show that based on SmI, greedy algorithm has performance guarantee for the maximization of possibly non-monotonic and non-submodular functions, justifying its usage for a much broader class of problems. We evaluate the theoretical results with several case studies, and also illustrate the application of the subset selection to causal structure learning.

## 1 Introduction

A wide variety of research disciplines, including computer science, economic, biology and social science, involve causality analysis of a network of interacting random processes. In particular, many of those tasks are closely related to subset selection. For example, in sensor network applications, with limited budget it is necessary to place sensors at information "sources" that provide the best observability of the system. To better predict a stock under consideration, investors need to select causal covariates from a pool of candidate information streams. We refer to the first type of problems as "causal sensor placement", and the second one as "causal covariate selection".

To solve the aforementioned problems we firstly need a causality measure for multiple random processes. In literature, there exists two types of causality definitions, one is related with time series prediction (called Granger-type causality) and another with counter-factuals [18]. We focus on Granger-type prediction causality substantiated with *Directed Information (DI)*, a tool from information theory. Recently, a large body of work has successfully employed DI in many research fields, including influence mining in gene networks [14], causal relationship inference in neural spike train recordings [19], and message transmission analysis in social media [23]. Compared to model-based or testing-based methods such as [2][21], DI is not limited by model assumptions and can naturally capture non-linear and non-stationary dependence among random processes. In addition, it has clear information theoretical interpretation and admits well-established estimation techniques. In this regards, we formulate causal sensor placement and covariate selection into cardinality constrained directed information maximizations problems.

We then need an efficient algorithm that makes optimal subset selection. Although subset selection, in general, is not tractable due to its combinatorial nature, the study of greedy heuristics for submodular objectives has shown promising results in both theory and practice. To list a few, following the pioneering work [8] that proves the near optimal $1 - 1/e$ guarantee, [12] [1] investigates the submod-

ularity of mutual information under Gaussian assumption, and then uses a greedy algorithm for sensor placement. In the context of speech and Nature Language Processing (NLP), the authors of and [13] adopt submodular objectives that encourage small vocabulary subset and broad coverage, and then proceed to maximization with a modified greedy method. In [3], the authors combine insights from spectral theory and submodularity analysis of $R^2$ score, and their result remarkably explains the near optimal performance of forward regression and orthogonal matching pursuit.

In this work, we also attack the causal subset selection problem via submodularity analysis. We show that the objective function of causal sensor placement, i.e., DI from selected set to its complement, is submodular, although not monotonic. And the problem of causal covariates selection, i.e., DI from selected set to some target process, is not submodular in general but is "nearly" submodular in particular cases. Since classic results require strictly submodularity and monotonicity which cannot be established for our purpose, we propose a novel measure of the degree of submodularity and show that, the performance guarantee of greedy algorithms can be obtained for possibly non-monotonic and non-submodular functions. Our contributions are: (1) Two important causal subset selection objectives are formulated with directed information and the corresponding submodularity analysis are conducted. (2) The SmI dependent performance bound implies that submodularity is better characterized by a continuous indicator than being used as a "yes or no" property, which extends the application of greedy algorithms to a much broader class of problems.

The rest of the paper is organized as follows. In next section, we briefly review the notion of directed information and submodular function. Section 3 is devoted to problem formulation and submodularity analysis. In section 4, we introduce SmI and provides theoretical results on performance guarantee of random and deterministic greedy algorithms. Finally in Section 5, we conduct experiments to justify our theoretical findings and illustrate a causal structure learning application.

## 2 Preliminary

**Directed Information** Consider two random process $X^n$ and $Y^n$, we use the convention that $X^i = \{X_0, X_1, ...X_i\}$, with $t = 0, 1, ..., n$ as the time index. Directed Information from $X^n$ to $Y^n$ is defined in terms of mutual information:

$$\mathcal{I}(X^n \to Y^n) = \sum_{t=1}^{n} I(X^t; Y_t | Y^{t-1}) \tag{1}$$

which can be viewed as the aggregated dependence between the history of process $X$ and the current value of process $Y$, given past observations of $Y$. The above definition captures a natural intuition about causal relationship, i.e., the unique information $X^t$ has on $Y_t$, when the past of $Y^{t-1}$ is known.

With *causally conditioned entropy* defined by $H(Y^n||X^n) \triangleq \sum_{t=1}^{n} H(Y_t|Y^{t-1}, X^t)$, the directed information from $X^n$ to $Y^n$ when *causally conditioned* on the series $Z^n$ can be written as

$$\mathcal{I}(X^n \to Y^n || Z^n) = H(Y^n||Z^n) - H(Y^n||X^n, Z^n) = \sum_{t=1}^{n} I(X^t; Y_t | Y^{t-1}, Z^t) \tag{2}$$

Observe that causally conditioned directed information is expressed as the difference between two causally conditioned entropy, which can be considered as "causal uncertainty reduction". With this interpretation one is able to relate directed information to Granger Causality. Denote $\bar{X}$ as the complement of $X$ in a a universal set $V$, then,

**Theorem 1** *[20] With log loss, $\mathcal{I}(X^n \to Y^n || \bar{X}^t)$ is precisely the value of the side information (expected cumulative reduction in loss) that $X$ has, when sequentially predicting $Y$ with the knowledge of $\bar{X}$. The predictors are distributions with minimal expected loss.*

In particular, with linear models directed information is equivalent to Granger causality for jointly Gaussian processes.

**Submodular Function** There are three equivalent definitions of submodular functions, and each of them reveals a distinct character of submodularity, a diminishing return property that universally exists in economics, game theory and network systems.

**Definition 1** *A submodular funciton is a set function $f : 2^\Omega \to \mathbb{R}$, which satisfies one of the three equivalent definitions:*
*(1) For every $S, T \subseteq \Omega$ with $S \subseteq T$, and every $x \in \Omega \setminus T$, we have that*

$$f(S \cup \{x\}) - f(S) \geq f(T \cup \{x\}) - f(T) \tag{3}$$

*(2) For every $S, T \subseteq \Omega$, we have that*

$$f(S) + f(T) \geq f(S \cup T) + f(S \cap T) \tag{4}$$

*(3) For every $S \subseteq \Omega$, and $x_1, x_2 \in \Omega \setminus S$, we have that*

$$f(S \cup \{x_1\}) + f(S \cup \{x_2\}) \geq f(S \cup \{x_1, x_2\}) + f(S) \tag{5}$$

A set function $f$ is called *supermodular* if $-f$ is submodular. The first definition is directly related to the diminishing return property. The second definition is better understood with the classic max $k$-cover problem [4]. The third definition indicates that the contribution of two elements is maximized when they are added individually to the base set. Throughout this paper, we will denote $f_x(S) \triangleq f(S \cup x) - f(S)$ as the "first order derivative" of $f$ at base set $S$ for further analysis.

## 3 Problem Formulation and Submodularity Analysis

In this section, we first formulate two typical subset selection problems into cardinality constrained directed information maximization. Then we address the issues of submodularity and monotonicity in details. All proofs involved in this and the other sections, are given in supplementary material.

**Causal Sensor Placement and Covariates Selection by Maximizing DI**     To motivate the first formulation, imagine we are interested in placing sensors to monitor pollution particles in a vast region. Ideally, we would like to put $k$ sensors, which is a given budget, at pollution sources to better predict the particle dynamics for other areas of interest. As such, the placement locations can be obtained by maximizing the directed information from selected location set $S$ to its complement $\bar{S}$ (in the universal set $V$ that contains all candidate sites). Then this type of "causal sensor placement" problems can be written as

$$\underset{S \subseteq V, |S| \leq k}{\operatorname{argmax}} \ \mathcal{I}(S^n \to \bar{S}^n) \tag{OPT1}$$

Regarding the causal covariates selection problem, the goal is to choose a subset $S$ from a universal set $V$, such that $S$ has maximal prediction causality to a (or several) target process $Y$. To leverage sparsity, the cardinality constraints $|S| \leq k$ is also imposed on the number of selected covariates. Again with directed information, this type of subset selection problems reads

$$\underset{S \subseteq V, |S| \leq k}{\operatorname{argmax}} \ \mathcal{I}(S^n \to Y^n) \tag{OPT2}$$

The above two optimizations are hard even in the most reduced cases: Consider a collection of causally independent Gaussian processes, then the above problems are equivalent to the D-optimal design problem, which has been shown to be NP-hard [11]. Unless "P = NP", it is unlikely to find any polynomial algorithm for the maximization, and a resort to tractable approximations is necessary.

**Submodularity Analysis of the Two Objectives**     Fortunately, we can show that the objective function of OPT1, the directed information from selected processes to unselected ones, is submodular.

**Theorem 2** *The objective $\mathcal{I}(S^n \to \bar{S}^n)$ as a function of $S \subseteq V$ is submodular.*

The problem is that OPT1 is *not monotonic* for all $S$, which can be seen since both $\mathcal{I}(\emptyset \to V)$ and $\mathcal{I}(V \to \emptyset)$ are 0 by definition. On the other hand, the deterministic greedy algorithm has guaranteed performance only when the objective function is monotonic up to $2k$ elements. In literature, several works have been addressing the issue of maximizing non-monotonic submodular function [6][7][17]. In this work we mainly analysis the random greedy technique proposed in [17], which is simpler compared to other alternatives and achieves best-known guarantees.

Concerning the second objective OPT2, we make a slight detour and take a look at the property of its "first derivative".

**Proposition 1** $f_x(S) = \mathcal{I}(S^n \cup x^n \to Y^n) - \mathcal{I}(S^n \to Y^n) = \mathcal{I}(x^n \to Y^n || S^n)$

Thus, the derivative is the directed information from processes $x$ to $Y$ causally conditioned on $S$. By the first definition of submodularity, if the derivative is decreasing in $S$, i.e. if $f_x(S) \geq f_x(T)$ for any $S \subseteq T \subseteq V$ and $x \subseteq V \setminus T$, then the objective $\mathcal{I}(S^n \to Y^n)$ is a submodular function. Intuition may suggest this is true since knowing more (conditioning on a larger set) seems to reduce the dependence (and also the causality) of two phenomena under consideration. However, in general, this conjecture is not correct, and a counterexample could be constructed by having "explaining away" processes. Hence the difficulty encountered for solved OPT2 is that the objective is *not submodular*. Note that with some extra conditional independence assumptions we can justify its submodularity,

**Proposition 2** *If for any two processes $s_1, s_2 \in S$, we have the conditional independence that $(s_{1t} \perp\!\!\!\perp s_{2t} \mid Y_t)$, then $\mathcal{I}(S^n \to Y^n)$ is a monotonic submodular function of set $S$.*

In practice, the assumption made in the above proposition is hard to check. Yet one may wonder that if the conditional dependence is weak or sparse, possibly a greedy selection still works to some extent because the submodularity is not severely deteriorated. Extending this idea we propose Submodularity Index (SmI), a novel measure of the degree of submodularity for general set functions, and we will provide the performance guarantee of greedy algorithms as a function of SmI.

## 4    Submodularity Index and Performance Guarantee

For the ease of notation, we use $f$ to denote a general set function and treat directed information objectives as special realizations. It's worth mentioning that in literature, several effort has already been made to characterize approximate submodularity, such as the $\varepsilon$ relaxation of definition (3) proposed in [5] for a dictionary selection objective, and the submodular ratio proposed in [3]. Compared to existing works, the SmI suggested in this work (1) is more generally defined for all set functions, (2) does not presume monotonicity, and (3) is more suitable for tasks involving information, influence, and coverage metrics in terms of computational convenience.

**SmI Definition and its Properties**    We start by defining the *local submodular index* for a function $f$ at location $A$ for a candidate set $S$

$$\varphi_f(S, A) \triangleq \sum_{x \in S} f_x(A) - f_S(A) \tag{6}$$

which can be considered as an extension of the third definition (5) of submodularity. In essence, it captures the *difference* between the sum of individual effect and aggregated effect on the first derivative of the function. Moreover, it has the following property:

**Proposition 3** *For a given submodular function $f$, the local submodular index $\varphi_f(S, A)$ is a supermodular function of $S$.*

Now we define SmI by minimizing over set variables:

**Definition 2** *For a set function $f : 2^V \to \mathbb{R}$ the submodularity index (SmI) for a location set $L$ and a cardinality $k$, denoted by $\lambda_f(L, k)$, is defined as*

$$\lambda_f(L, k) \triangleq \min_{\substack{A \subseteq L \\ S \cap A = \emptyset, \, |S| \leq k}} \varphi_f(S, A) \tag{7}$$

Thus, SmI is the smallest possible value of local submodular indexes subject to $|S| \leq k$. Note that we implicitly assume $|S| \geq 2$ in the above definition, as in the cases where $|S| = \{0, 1\}$, SmI reduces to $0$. Besides, the definition of submodularity can be alternatively posed with SmI,

**Lemma 1** *A set function $f$ is submodular if and only if $\lambda_f(L, k) \geq 0$, $\forall L \subseteq V$ and $k$.*

For functions that are already submodular, SmI measures how strong the submodularity is. We call a function *super-submodular* if its SmI is strictly larger than zero. On the other hand for functions that are not submodular, SmI provides an indicator of how close the function is to submodular. We call a function *quasi-submodular* if it has a negative but close to zero SmI.

Direct computation of SmI by solving (7) is hard. For the purpose of obtaining performance guarantee, however, a lower bound of SmI is sufficient and is much easier to compute. Consider the objective of (OPT1), which is already a submodular function. By using proposition 3, we conclude that its local submodular index is a super-modular function for fixed location set. Hence computing (7) becomes a cardinality constrained supermodular minimization problem for each location set. Besides, the following decomposition is useful to avoid extra work on directed information estimation:

**Proposition 4** *The local submodular index of the function $\mathcal{I}(\{\bullet\}^n \to \{V \setminus \bullet\}^n)$ can be decomposed as $\varphi_{\mathcal{I}(\{\bullet\}^n \to \{V \setminus \bullet\}^n)}(S^n, A^n) = \varphi_{H(\{V \setminus \bullet\}^n)}(S^n, A^n) + \sum_{t=1}^{n} \varphi_{H(\{\bullet\}|V^{t-1})}(S_t, A_t)$, where $H(\bullet)$ is the entropy function.*

The lower bound of SmI for the objective of OPT2 is more involved. With some work on an alternative representation of causally conditioned directed information, we obtain that

**Lemma 2** *For any location sets $L \subseteq V$, cardinality $k$, and target process set $Y$, we have*

$$\lambda_{\mathcal{I}(\{\bullet\}^n \to Y^n)}(L, k) \geq \min_{\substack{W \subseteq V \\ |W| \leq |\overline{L}| + k}} \sum_{t=1}^{n} \left\{ \mathcal{G}_{|L|+k}\left(W^t, Y^{t-1}\right) - \mathcal{G}_{|L|+k}\left(W^t, Y^t\right) \right\} \tag{8}$$

$$\geq - \max_{\substack{W \subseteq V \\ |W| \leq |\overline{L}| + k}} \mathcal{I}(W^n \to Y^n) \geq -\mathcal{I}(V^n \to Y^n) \tag{9}$$

*where the function $\mathcal{G}_k(W, Z) \triangleq \sum_{w \in W} H(w|Z) - kH(W|Z)$ is super-modular of $W$.*

Since (8) is in fact minimizing (maximizing) the difference of two supermodular (submodular) functions, one can use existing approximate or exact algorithms [10] [16] to compute the lower bound. (9) is often a weak lower bound, although is much easier to compute.

**Random Greedy Algorithm and Performance Bound with SmI**     With the introduction of SmI, in this subsection, we analyze the performance of the random greedy algorithm for maximizing non-monotonic, quasi- or super-submodular function in a unified framework. The results broaden the theoretical guarantee for a much richer class of functions.

---

**Algorithm 1** Random Greedy for Subset Selection

> $S_0 \leftarrow \phi$
> **for** $i = 1, ..., k$ **do**
>     $M_i = \text{argmax}_{M_i \subseteq V \setminus S_{i-1}, |M_i| = k} \sum_{u \in M_i} f_u(S_i)$
>
>     Draw $u_i$ uniformly from $M_i$
>     $S_i \leftarrow S_{i-1} \cup \{u_i\}$
> **end for**

---

The randomized greedy algorithm was recently proposed in [17] [22] for maximizing cardinality constrained *non-monotonic submodular* functions. Also in [17], a $1/e$ expected performance bound was provided. The overall procedure is summarized in algorithm 1 for reference. Note that the random greedy algorithm only requires $O(k|V|)$ calls of the function evaluation, making it suitable for large-scale problems.

In order to analyze the performance of the algorithm, we start with two lemmas that reveal more properties of SmI. The first lemma shows that the monotonicity of the first derivative of a general set function $f$ could be controlled by its SmI.

**Lemma 3** *Given a set function $f : V \to \mathbb{R}$, and the corresponding SmI $\lambda_f(L, k)$ defined in (7), and also let set $B = A \cup \{y_1, ..., y_M\}$ and $x \in \overline{B}$. For an ordering $\{j_1, ..., j_M\}$, define $B_m = A \cup \{y_{j_1}, ..., y_{j_m}\}$, $B_0 = A$, $B_M = B$, we have*

$$f_x(A) - f_x(B) \geq \max_{\{j_1, ..., j_M\}} \sum_{m=0}^{M-1} \lambda_f(B_m, 2) \geq M\lambda_f(B, 2) \tag{10}$$

Essentially, the above result implies that as long as SmI can be lower bounded by some small negative number, the submodularity (the decreasing derivative property (3) in Definition 1) is not severely degraded. The second lemma provides an SmI dependent bound on the expected value of a function with random arguments.

**Lemma 4** *Let the set function $f : V \to \mathbb{R}$ be quasi submodular with $\lambda_f(L, k) \leq 0$. Also let $S(p)$ a random subset of $S$, with each element appears in $S(p)$ with probability at most $p$, then we have $E[f(S(p))] \geq (1 - p_1)f(\emptyset) + \gamma_{S,p}$, with $\gamma_{S,p} \triangleq \sum_{i=1}^{|S|} (i-1)p\lambda_f(S_i, 2)$.*

Now we present the main theory and provide refined bounds for two different cases when the function is monotonic (but not necessarily submodular) or submodular (but not necessarily monotonic).

**Theorem 3** *For a general (possibly non-monotonic, non-submodular) functions $f$, let the optimal solution of the cardinality constrained maximization be denoted as $S^*$, and the solution of random greedy algorithm be $S^g$ then*

$$E\left[f(S^g)\right] \geq \left(\frac{1}{e} + \frac{\xi_{S^g,k}^f}{E[f(S^g)]}\right) f(S^*)$$

*where $\xi_{S^g,k}^f = \lambda_f(S_g,k) + \frac{k(k-1)}{2}\min\{\lambda_f(S_g,2),0\}$.*

The role of SmI in determining the performance of the random greedy algorithm is revealed: the bound consist of $1/e \approx 0.3679$ plus a term as a function of SmI. If $SmI = 0$, the $1/e$ bound in previous literature is recovered. For super-submodular functions, as SmI is strictly larger than zero, the theorem provides a stronger guarantee by incorporating SmI. For quasi-submodular functions having negative SmI, although a degraded guarantee is produced, the bound is only slightly deteriorated when SmI is close to zero. In short, the above theorem not only encompasses existing results as special cases, but also suggests that we should view submodularity and monotonicity as a "continuous" property of set functions. Besides, greedy heuristics should not be restricted to the maximization of submodular functions, but can also be applied for "quasi-submodular" functions because a near optimal solution is still achievable theoretically. As such, we can formally define quasi-submodular functions as those having an SmI such that $\frac{\xi_{S,k}^f}{E[f(S)]} > -\frac{1}{e}$.

**Corollary 1** *For monotonic functions in general, random greedy algorithm achieves*

$$E\left[f(S^g)\right] \geq \left(1 - \frac{1}{e} + \frac{\lambda_f'(S^g,k)}{E\left[f(S^g)\right]}\right) f(S^*)$$

*and deterministic greedy algorithm also achieves $f(S^g) \geq \left(1 - \frac{1}{e} + \frac{\lambda_f'(S^g,k)}{f(S^g)}\right) f(S^*)$, where*
$\lambda_f'(S^g,k) = \begin{cases} \lambda_f(S^g,k) & if \quad \lambda_f(S^g,k) < 0 \\ (1-1/e)^2\lambda_f(S^g,k) & if \quad \lambda_f(S^g,k) \geq 0 \end{cases}$.

We see that in the monotonic case, we get a stronger bound for submodular functions compared to the $1 - 1/e \approx 0.6321$ guarantee. Similarly, for quasi-submodular functions, the guarantee is degraded but not too much if SmD is close to 0. Note that the objective function of OPT2 fits into this category. For submodular but non-monotonic functions, e.g., the objective function of OPT1, we have

**Corollary 2** *For submodular function that are not necessarily monotonic, random greedy algorithm has performance*

$$E\left[f(S^g)\right] \geq \left(\frac{1}{e} + \frac{\lambda_f(S^g,k)}{E\left[f(S^g)\right]}\right) f(S^*)$$

## 5  Experiment and Applications

In this section, we conduct experiments to verify the theoretical results, and provide an example that uses subset selection for causal structure learning.

**Data and Setup**    The synthesis data is generated with the Bayes network Toolbox (BNT) [15] using dynamic Bayesian network models. Two sets of data, denoted by *D1* and *D2*, are simulated, each containing 15 and 35 processes, respectively. For simplicity, all processes are $\{0,1\}$ valued. The processes are created with both simultaneous and historical dependence on each other. The order (memory length) of the historical dependence is set to 3. The MCMC sampling engine is used to draw $n = 10^4$ points for both D1 and D2. The stock market dataset, denoted by *ST*, contains hourly values of 41 stocks and indexes for the years 2014-2015. Note that data imputation is performed to amend a few missing values, and all processes are aligned in time. Moreover, we detrend each time

series with a recursive HP-filter [24] to remove long-term daily or monthly seasonalities that are not relevant for hourly analysis.

Directed information is estimated with the procedure proposed in [9], which adopts the context tree weighting algorithm as an intermediate step to learn universal probability assignment. Interested readers are referred to [19][20] for other possible estimators. The maximal context tree depth is set to 5, which is sufficient for both the synthesis datasets and the real-world ST dataset.

**Causal Subset Selection Results**

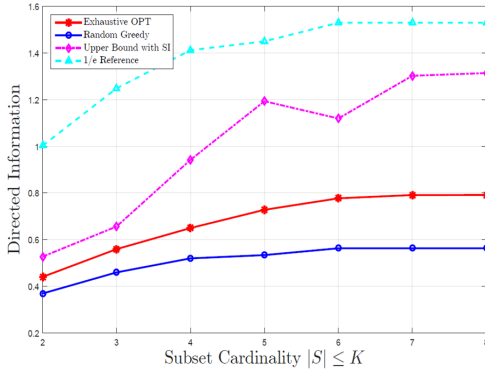 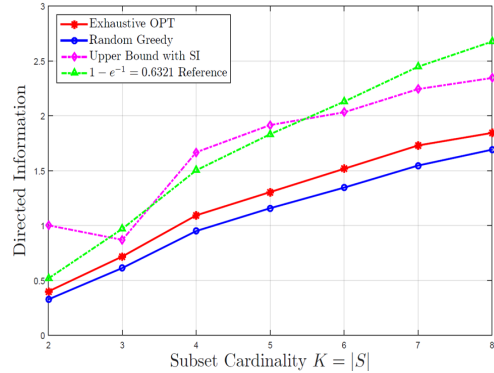

Figure 1: Solution and Bounds for OPT1 on D1    Figure 2: Solution and Bounds for OPT2 on ST

Firstly, the causal sensor placement problem, OPT1, is solved on data set D1 with the random greedy algorithm. Figure 1 shows the optimal solution by exhaustive search (red-star), random greedy solution (blue-circle), the $1/e$ reference bound (cyan-triangle), and the bound with SmI (magenta-diamond), each for cardinality constraints imposed from $k = 2$ to $k = 8$. It is seen that the random greedy solution is close to the true optimum. In terms of computational time, the greedy method finishes in less than five minutes, while the exhaustive search takes about 10 hours on this small-scale problem ($|V| = 15$). Comparing two bounds in Figure 1, we see that the theoretical guarantee is greatly improved, and a much tighter bound is produced with SmI. The corresponding normalized SmI values, defined by $\frac{SmI}{f(L^g)}$, is shown in the first row of Table 1. As a consequence of those strictly positive SmI values and Corollary 2, the guarantees are made greater than $1/e$. This observation justifies that the bounds with SmI are better indicators of the performance of the greedy heuristic.

Table 1: Normalized submodularity index (NSmI) for OPT1 on D1 and OPT2 on ST at locations of greedy selections. Cardinality is imposed from $k = 2$ to $k = 8$.

| $k =$ | 2 | 3 | 4 | 5 | 6 | 7 | 8 |
|---|---|---|---|---|---|---|---|
| normalized SmI for OPT1 | 0.382 | 0.284 | 0.175 | 0.082 | 0.141 | 0.078 | 0.074 |
| normalized SmI for OPT2 | -0.305 | 0.071 | -0.068 | -0.029 | 0.030 | 0.058 | 0.092 |

Secondly, the causal covariates selection problem, OPT2, is solved on ST dataset with the stock XOM used as the target process $Y$. The results of random greedy, exhaustive search, and performance bound (Corollary 1) are shown in Figure 2, and normalized SmIs are listed in the second row of Table 1. Note that the $1 - 1/e$ reference line (green-triangle) in the figure is only for comparison purpose and is NOT an established bound. We observe that although the objective is not submodular, the random greedy algorithm is still near optimal. As we compare the reference line and the bound calculated with SmI (magenta-diamond), we see that the performance guarantee can be either larger or smaller than $1 - 1/e$, depending on the sign of SmI. By definition, SmI measures the submodularity of a function at a location set. Hence, the SmI computed at each greedy selection captures the "local" submodularity of the function. The central insight gained from this experiment is that, for a function lacking general submodularity, such as the objective function of OPT2, it can be quasi-submodular ($SmI \leq 0$, $SmI \approx 0$) or super-submodular ($SmI > 0$) at different locations. Accordingly the performance guarantee can be either larger or smaller than $1 - 1/e$, depending on the values of SmI.

**Application: Causal Structure Learning**    The greedy method for subset selection can be used in many situations. Here we briefly illustrate the structure learning application based on covariates selection. As is detailed in the supplementary material and [20], one can show that the causal structure learning problem can be reduces to solving $\operatorname{argmax}_{S \subseteq V, |S| \leq k} \mathcal{I}(S^n \to X_i^n)$ for each node $i \in V$,

assuming maximal in degree is bounded by $k$ for all nodes. Since the above problem is exactly the covariate selection considered in this work, we can reconstruct the causal structure for a network of random processes by simply using the greedy heuristic for each node.

Figure 3 and Figure 4 illustrate the structure learning results on D1 and D2, respectively. In both two figures, the left subfigure is the ground truth structure, i.e., the dynamic Bayesian networks that are used in the data generation. Note that each node in the figure represents a random process, and an edge from node $i$ to $j$ indicates a causal (including both simultaneous and historical) influence. The subfigure on the right shows the reconstructed causal graph. Comparing two subfigures in Figure 3, we observe that the simple structure learning method performs almost flawlessly. In fact, only the edge $6 \rightarrow 4$ is miss detected. On the larger case D2 with 35 processes, the method still works relatively well, correctly reconstructing 82.69% causal relations. Given that only the maximal in degree for all nodes is assumed a priori, these results not only justify the greedy approximation for the subset selection, but also demonstrate its effectiveness in causal structure learning applications.

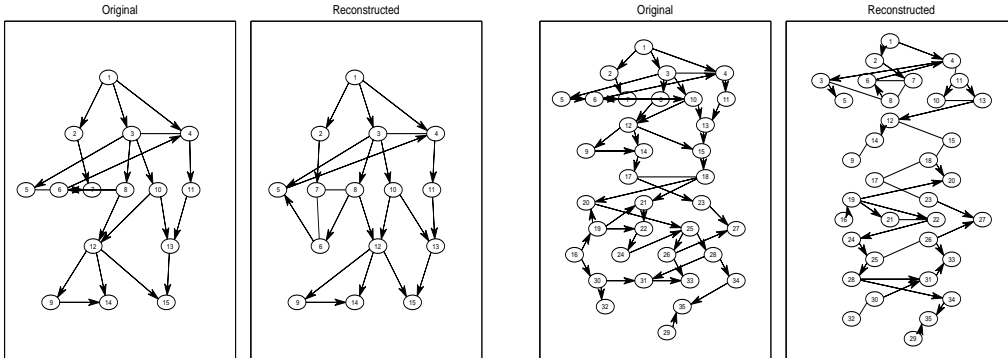

Figure 3: Ground truth structure (left) versus Reconstructed causal graph (right), D1 dataset

Figure 4: Ground truth structure (left) versus Reconstructed causal graph (right), D2 dataset

# 6   Conclusion

Motivated by the problems of source detection and causal covariate selection, we start with two formulations of directed information based subset selection, and then we provide detailed submodularity analysis for both of the objective functions. To extend the greedy heuristics to possibly non-monotonic, approximately submodular functions, we introduce an novel notion, namely submodularity index, to characterize the "degree" of submodularity for general set functions. More importantly, we show that with SmI, the theoretical performance guarantee of greedy heuristic can be naturally extended to a much broader class of problems. We also point out several bounds and techniques that can be used to calculate SmI efficiently for the objectives under consideration. Experimental results on the synthesis and real data sets reaffirmed our theoretical findings, and also demonstrated the effectiveness of solving subset selection for learning causal structures.

# 7   Acknowledgments

This research is funded by the Republic of Singapore's National Research Foundation through a grant to the Berkeley Education Alliance for Research in Singapore (BEARS) for the Singapore-Berkeley Building Efficiency and Sustainability in the Tropics (SinBerBEST) Program. BEARS has been established by the University of California, Berkeley as a center for intellectual excellence in research and education in Singapore. We also thank the reviews for their helpful suggestions.

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
