[Supplementary Material · suproof.pdf]



# Supplementary Material: Causal Meet Submodular: Subset Selection with Directed Information

**Yuxun Zhou**                                                                    YXZHOU@CS.BERKELEY.EDU
*Department of EECS, University of California, Berkeley*
*Berkeley,CA 94720-1776, USA*

## Abstract

This is the supplementary material for the paper entitled "Causal Meet Submodular: Subset Selection with Directed Information" presented in NIPS 2016. Proofs of main theorems and lemmas are given in details. We also provide additional numerical experiment which does fit in the paper due to the page limit.

## 1. Proofs

**Theorem 2** *The objective* $\mathcal{I}(A^n \to \bar{A}^n)$ *as a function of* $A \subseteq V$ *is submodular.*

**Proof** Let's first show a property of mutual information. At time $t$, we have

$$I\left(A^t \cup \{y\}^t; \overline{A \cup \{y\}}_t | \overline{A \cup \{y\}}^{t-1}\right) - I\left(A^t; \bar{A}_t | \bar{A}^{t-1}\right)$$

$$= H(V^{t-1}, A_t, y_t) + H\left(\overline{A \cup \{y\}}^t\right) - H(V^t) - H\left(\overline{A \cup \{y\}}^{t-1}\right)$$

$$\quad - H(V^{t-1}, A_t) - H(\bar{A}^t) + H(V^t) + H(\bar{A}^{t-1})$$

$$= H(y_t | V^{t-1}, A_t) - H\left(y^t | \overline{A \cup \{y\}}^t\right) + H\left(y^{t-1} | \overline{A \cup \{y\}}^{t-1}\right)$$

where we use $I(X, Y|Z) = H(X, Z) + H(Y, Z) - H(X, Y, Z) - H(Z)$. Summing over the last formula over $t$ and canceling telescoping terms, we obtain the following formula by the definition of directed information,

$$\mathcal{I}\left(A^n \cup \{y^n\} \to \overline{A \cup \{y\}}^n\right) - \mathcal{I}\left(A^n \to \bar{A}^n\right)$$

$$= \sum_t H(y_t | V^{t-1}, A_t) - H\left(y^n | \overline{A \cup \{y\}}^n\right) + H\left(y_0\right)$$

where we assumed independent initial distribution.[1] Now for any set $B \supseteq A$, by "information never hurt"

$$H(y_t | V^{t-1}, A_t) \geq H(y_t | V^{t-1}, B_t)$$

$$H\left(y^n | \overline{A \cup \{y\}}^n\right) \leq H\left(y^n | \overline{B \cup \{y\}}^n\right)$$

Hence Definition 1 of submodularity is verified. The objective function is submodular. ∎

**Proposition 1** $f_X(S) = \mathcal{I}(S^n \cup x^n \to Y^n) - \mathcal{I}(S^n \to Y^n) = \mathcal{I}(x^n \to Y^n || S^n)$

---

1. In fact, initial condition does not matter for large $t$, which is usually true for meaningful DI estimation

**Proof** Note the following alternative expression for DI:

$$
\begin{aligned}
\mathcal{I}(X^n \to Y^n) &= \sum_{t=1}^{n} \left\{ H(Y_t|Y^{t-1}) - H(Y_t|Y^{t-1}, X^t) \right\} \\
&= H(Y^n) - \sum_{t=1}^{n} H(Y_t|Y^{t-1}, X^t)
\end{aligned}
\tag{1}
$$

and the result can be obtained since $H(Y^n||X^n) \triangleq \sum_{t=1}^{n} H(Y_t|Y^{t-1}, X^t)$, and the directed information from $X^n$ to $Y^n$ when *causally conditioned* on the series $Z^n$ can be written as

$$
\mathcal{I}(X^n \to Y^n||Z^n) = H(Y^n||Z^n) - H(Y^n||X^n, Z^n) = \sum_{t=1}^{n} I(X^t; Y_t|Y^{t-1}, Z^t)
\tag{2}
$$

∎

**Proposition 2** *If for any two processes $s_1, s_2 \in S$, we have the conditional independence that $(s_{1t} \perp\!\!\!\perp s_{2t} \mid Y_t)$, then $\mathcal{I}(S^n \to Y^n)$ is a monotonic submodular function of set $S$.*

**Proof** In this case, we see that the probabilistic model reduces to "causal naive Bayesian", and the submodulaity follows by check Definition 1 with conditional independence and Proposition 1. ∎

**Lemma 1** *A set function $f$ is submodular if and only if $\lambda_f(L, k) \geq 0$, $\forall L \subseteq V$ and $k$.*

**Proof** Simply take $k = 2$, then $\lambda_{V,2} \geq 0$ implies definition 3 of submodularity, hence $f$ is submodular. For the other direction, assuming $f$ is submodular, then for any $A, S \subseteq V$ and $x_i \in S$ by telescoping

$$
\begin{aligned}
f(A \cup S) - f(A) &= \sum_{i=1}^{|S|} f(A \cup S_{(i)} \cup x_i) - f(A \cup S_{(i)}) \\
&\leq \sum_{i=1}^{|S|} [f(A \cup x_i) - f(A)]
\end{aligned}
$$

where $S_{(i)} \triangleq S \setminus \{x_1, ..., x_i\}$ and with the definition of submodularity deviance we get $\lambda_{V,k} \geq 0$ ∎

**Lemma 2** *For any location sets $L \subseteq V$, cardinality $k$, and target process set $Y$, we have*

$$
\lambda_{\mathcal{I}(\{\bullet\}^n \to Y^n)}(L, k) \geq \min_{\substack{W \subseteq V \\ |W| \leq |L| + k}} \sum_{t=1}^{n} \left\{ \mathcal{G}_{|L|+k}\left(W^t, Y^{t-1}\right) - \mathcal{G}_{|L|+k}\left(W^t, Y^t\right) \right\}
\tag{3}
$$

$$
\geq - \max_{\substack{W \subseteq V \\ |W| \leq |L| + k}} \mathcal{I}(W^n \to Y^n) \geq -\mathcal{I}(V^n \to Y^n)
\tag{4}
$$

where the function $\mathcal{G}_k(W, Z) \triangleq \sum_{w \in W} H(w|Z) - kH(W|Z)$ defined in terms of entropy is super-modular of $W$.

**Proof** First note that for any random variable set $U$, we have

$$
\begin{aligned}
&\mathcal{I}(U^n \to Y^n || A^n) \\
&= H(Y^n || A^n) - H(Y^n || U^n, A^n) \\
&= \sum_{t=1}^{n} I(U^t; Y_t | A^t, Y^{t-1}) \\
&= \sum_{t=1}^{n} \left\{ H(U^t | A^t, Y^{t-1}) - H(U^t | A^t, Y^t) \right\}
\end{aligned}
$$

Hence by plugging in with $x^t$, $S^t$ and rearrange, we get

$$
\begin{aligned}
\lambda_{Y^n, A^n}(S^n) &= \sum_{t=1}^{n} \left\{ \sum_{x \in S} H(x^t | A^t, Y^{t-1}) - H(S^t | A^t, Y^{t-1}) \right. \\
&\qquad\qquad \left. - \left[ \sum_{x \in S} H(x^t | A^t, Y^t) - H(S^t | A^t, Y^t) \right] \right\} \\
&= \sum_{t=1}^{n} \left\{ \mathcal{G}_{A^t, Y^{t-1}}(S^t) - \mathcal{G}_{A^t, Y^t}(S^t) \right\}
\end{aligned}
$$

Let's verify several properties of $\mathcal{G}$
• $\mathcal{G}$ is Supermodular
Remember "information never hurts" inequality, we get

$$
\begin{aligned}
\mathcal{G}_k(W \cup \{y\}, Z) - \mathcal{G}_k(W, Z) &= H(y|Z) - kH(y|W, Z) \\
&\leq H(y|Z) - kH(y|L, Z)
\end{aligned}
$$

for $W \subseteq L$. Hence by definition $\mathcal{G}_k(W, Z)$ is supermodular.

$$
\begin{aligned}
&\mathcal{G}_k(W, Z_1) - \mathcal{G}_k(W, Z_2) \\
&= \sum_{w \in W} [H(w|Z_1) - H(w|Z_2)] - k [H(W|Z_1) - H(W|Z_2)]
\end{aligned}
$$

is decreasing in $k$ as $H(W|Z_1) \geq H(W|Z_2)$ for $Z_1 \subseteq Z_2$
• $\mathcal{G}$ is Posimodular Recall that a set function is posimodular iif

$$
f(S) + f(T) \geq f(S \setminus T) + f(T \setminus S)
$$

Let's check

$$\mathcal{G}_1(S, Z) + \mathcal{G}_1(T, Z) - \mathcal{G}_1(S \setminus T, Z) - \mathcal{G}_1(T \setminus S, Z)$$

$$= \sum_{x \in S} H(x|Z) + \sum_{x \in T} H(x|Z) - H(S|Z) - H(T|Z)$$

$$\quad - \sum_{x \in S \setminus T} H(x|Z) - \sum_{x \in T \setminus S} H(x|Z)$$

$$\quad - H(S \setminus T|Z) - H(T \setminus S|Z)$$

$$= 2 \sum_{x \in S \cap T} H(x|Z) - H(S \cap T|S \setminus T, Z) - H(S \cap T|T \setminus S, Z)$$

$$\geq 2 \sum_{x \in S \cap T} H(x|Z) - 2H(S \cap T|Z) \geq 0$$

The last inequality is due to submodularity of $H(\bullet|Z)$ Now let's proof the lemma. Since $H(x|A, Y) = H(x|Y) - H(A|Y) + H(A|x, Y)$, and for any $x \in A$, $H(x|A, Y) = 0$, we have

$$\mathcal{G}_1\left(S^t, \{A^t, Y^{t-1}\}\right)$$

$$= \sum_{x \in S} H(x^t|A^t, Y^{t-1}) - H(S^t|A^t, Y^{t-1})$$

$$= \sum_{x \in S \cup A} H(x^t|A^t, Y^{t-1}) - H(S^t|A^t, Y^{t-1})$$

$$= \sum_{x \in S \cup A} \left\{ H(x^t|Y^{t-1}) - H(A^t|Y^{t-1}) + H(A^t|x^t, Y^{t-1}) \right\} - H(S^t \cup A^t|Y^{t-1}) + H(A^t|Y^{t-1})$$

$$= \underbrace{\sum_{x \in S \cup A} H(x^t|Y^{t-1}) - H(S^t \cup A^t|Y^{t-1})}_{\mathcal{G}_1(S^t \cup A^t, \{Y^{t-1}\})} - \sum_{x \in S \cup A} H(A^t|Y^{t-1}) + \sum_{x \in S \cup A} H(A^t|x^t, Y^{t-1}) + H(A^t|Y^{t-1})$$

Similar equality could be derived for $\mathcal{G}_1\left(S^t, \{A^t, Y^t\}\right)$, then their difference

$$\mathcal{G}_1\left(S^t, \{A^t, Y^{t-1}\}\right) - \mathcal{G}_1\left(S^t, \{A^t, Y^t\}\right)$$

$$= \mathcal{G}_1\left(S^t \cup A^t, \{Y^{t-1}\}\right) - \mathcal{G}_1\left(S^t \cup A^t, \{Y^t\}\right) + H(A^t|Y^{t-1}) - H(A^t|Y^t)$$

$$\quad + \sum_{x \in S \cup A} \left[ H(A^t|x^t, Y^{t-1}) - H(A^t|x^t, Y^t) \right] - \sum_{x \in S \cup A} \left[ H(A^t|Y^{t-1}) - H(A^t|Y^t) \right] \qquad (5)$$

Now note that $H(A^t|Y^{t-1}) - H(A^t|Y^t) = I(A^t; Y_t|Y^{t-1})$ and $H(A^t|x^t, Y^{t-1}) - H(A^t|x^t Y^t) = I(A^t; Y_t|x^t, Y^{t-1})$ are both positive and increasing in $A$. We get

$$- \sum_{x \in S \cup A} \left[ H(A^t|Y^{t-1}) - H(A^t|Y^t) \right]$$

$$\geq - \sum_{x \in S \cup A} \left[ H(A^t \cup x^t|Y^{t-1}) - H(A^t \cup x^t|Y^t) \right]$$

$$= - \sum_{x \in S \cup A} \left[ H(A^t|x^t, Y^{t-1}) - H(A^t|x^t, Y^t) \right] - \sum_{x \in S \cup A} \left[ H(x^t|Y^{t-1}) - H(x^t|Y^t) \right]$$

Plug into (5) and cancel terms, we get

$$\mathcal{G}_1\left(S^t, \{A^t, Y^{t-1}\}\right) - \mathcal{G}_1\left(S^t, \{A^t, Y^t\}\right)$$
$$\geq -\left[H(A^t \cup S^t | Y^{t-1}) - H(A^t \cup S^t | Y^t)\right]$$
$$= -I(A^t \cup S^t; Y_t | Y^{t-1})$$

On the other hand, if we relax the third term in (5) and use the increasing property of $I(A^t; Y_t | Y^{t-1})$

$$\mathcal{G}_1\left(S^t, \{A^t, Y^{t-1}\}\right) - \mathcal{G}_1\left(S^t, \{A^t, Y^t\}\right)$$
$$\geq \mathcal{G}_1\left(S^t \cup A^t, \{Y^{t-1}\}\right) - \mathcal{G}_1\left(S^t \cup A^t, \{Y^t\}\right)$$
$$- (|S \cup A| - 1)\left[H(A^t \cup S^t | Y^{t-1}) - H(A^t \cup S^t | Y^t)\right]$$
$$= \mathcal{G}_{|S \cup A|}\left(S^t \cup A^t, \{Y^{t-1}\}\right) - \mathcal{G}_{|S \cup A|}\left(S^t \cup A^t, \{Y^t\}\right)$$
$$\geq \mathcal{G}_{|L|+k}\left(S^t \cup A^t, \{Y^{t-1}\}\right) - \mathcal{G}_{|L|+k}\left(S^t \cup A^t, \{Y^t\}\right)$$

as $|L| + k \geq |S \cup A|$ and the second properties of function $\mathcal{G}$. Now the inequalities follows from the definition of directed information and the fact that for any $S, A \subseteq V$ that satisfies $A \subseteq L$, $S \cap A = \emptyset$, $|S| \leq k$, they are also feasible solutions for $W = S \cup A : |S \cup A| \leq |L| + k$. ∎

**Lemma 3** *Given a set function $f : V \to \mathbb{R}$, the corresponding SmI $\lambda_f(L, k)$, and also let set $B = A \cup \{y_1, ..., y_M\}$ and $x \in \bar{B}$. For an ordering $\{j_1, ..., j_M\}$, define $B_m = A \cup \{y_{j_1}, ..., y_{j_m}\}$, $B_0 = A$, $B_M = B$, we have*

$$f_x(A) - f_x(B) \geq \max_{\{j_1,...,j_M\}} \sum_{m=0}^{M-1} \lambda_f(B_m, 2) \geq M\lambda_f(B, 2) \tag{6}$$

**Proof** Let $k = 1$, $S = \{x_1, x_2\}$ and by our definition of SmI

$$\sum_{x \in S} f(A \cup x) - f(A) - [f(A \cup S) - f(A)] \geq \lambda_{A,2}$$

Rearranging gives

$$f(A \cup x_1) - f(A) - [f(A \cup x_1 \cup x_2) - f(A \cup x_2)] \geq \lambda_{A,2}$$

or with the notation of derivative

$$f_{x_1}(A) - f_{x_1}(A \cup x_2) \geq \lambda_{A,2} \tag{7}$$

This is somewhat a "trimming" property. Now consider $A \subseteq B \subseteq V$. Let's write explicitly $B_j = A \cup \{y_1, ..., y_j\}$, $B_0 = A$, $B_m = B$ with $m = |B| - |A|$, then

$$f_x(B_j) \leq f_x(B_{j-1}) - \lambda_{B_{j-1},2}$$

for $j = 1, ..., m$. Adding the $m$ equations we get

$$f(A) - f(B) \geq \sum_{j=1}^{|B|-|A|} \lambda_{B_j,2} \tag{8}$$

Also note that the order of $y_1, ..., y_m$ does not matter. Hence the proposition. ∎

**Lemma 4** *Let the set function $f : V \to \mathbb{R}$ be quasi submodular with $\lambda_f(L, k) \leq 0$. Also let $S(p)$ a random subset of $S$, with each element appears in $S(p)$ with probability at most $p$, then*

$$E\left[f(S(p))\right] \geq (1 - p_1)f(\emptyset) + \gamma_{S,p}$$

*with $\gamma_{S,p} \triangleq \sum_{i=1}^{|S|}(i-1)p\lambda_f(S_i, 2)$*

**Proof** W.l.o.g. assume elements in $S$ are ordered by its probability to be in $S(p)$, i.e. $S = \{u_1, u_2, ..., u_{|S|}\}$ and $p_i = \mathbb{P}(u_i \in S(p)) \geq P(u_j \in S(p)) = p_j$ for any $1 \leq i \leq j \leq |S|$. Define $S_i = \{u_1, u_2, ..., u_i\}$, $S_0 = \emptyset$. Then

$$E\left[f(S(p))\right] = E\left[f(\emptyset) + \sum_{i=1}^{|S|} \mathbb{I}_{\{u_i \in S(p)\}} f_{u_i}(S_{i-1} \cap S(p))\right]$$

$$\geq E\left[f(\emptyset) + \sum_{i=1}^{|S|} \mathbb{I}_{\{u_i \in S(p)\}}\left[f_{u_i}(S_{i-1}) + (i-1)\lambda_{S_{i-1},2}\right]\right]$$

$$= f(\emptyset) + \sum_{i=1}^{|S|}\left[p_i f_{u_i}(S_{i-1}) + (i-1)p_i\lambda_{S_{i-1},2}\right]$$

$$= (1 - p_1)f(\emptyset) + \sum_{i=1}^{|S|}(p_{i-1} - p_i)f(S_i) + p_{|S|}f(S) + \sum_{i=1}^{|S|}(i-1)p_i\lambda_{S_i,2}$$

$$\geq (1 - p_1)f(\emptyset) + \sum_{i=1}^{|S|}(i-1)p_i\lambda_{S_i,2}$$

$$= (1 - p_1)f(\emptyset) + \gamma_{S,p}$$

where the first inequality is due to last proposition, and second inequality is a direct result of the assumption that $p_i$'s are in decreasing order. Now if $f$ is strongly submodular, then by the definition of $\lambda_{S,k}$, we see that $\gamma_{S,p} \geq 0$, otherwise if $f$ is only approximately submodular with $\lambda_{S,k} \leq 0$, we have

$$\gamma_{S,p} \geq \sum_{i=1}^{|S|}(i-1)p_1\lambda_{S,2} \geq \frac{|S|(|S|-1)}{2}\lambda_{S,2} \triangleq \beta_S$$

∎

**Theorem 3** *For a general (non-monotonic, non-submodular) functions $f$, let the optimal solution of the cardinality constrained maximization be denoted as $S^*$, and the solution of random greedy algorithm be $S^g$ then*

$$E\left[f(S^g)\right] \geq \left(\frac{1}{e} + \frac{\xi^f_{S^g,k}}{E[f(S^g)]}\right) f(S^*)$$

*where $\xi^f_{S^g,k} = \lambda_f(S_g, k) + \frac{k(k-1)}{2} \min\{\lambda_f(S_g, 2), 0\}$*

**Proof** Let $\mathcal{C}^i$ be the event of random choices up to iteration $i$ according to the algorithm. Then by tower property

$$E\left[f_{x_{i+1}}(S_i)\right] = E\left[E\left[f_{x_{i+1}}(S_i)|\mathcal{C}^i\right]\right]$$

Denote $S^*$ the true optimal. The inside expectation is just

$$
\begin{aligned}
E\left[f_{x_{i+1}}(S_i)|\mathcal{C}^i\right] &= \frac{1}{k} \sum_{x \in M_{i+1}} f_x(S_i) \geq \frac{1}{k} \sum_{x \in S^* \setminus S_i} f_x(S_i) \\
&\geq \frac{1}{k}\left[\lambda_{S_i,|S^* \setminus S_i|} + f(S^* \cup S_i) - f(S_i)\right]
\end{aligned}
\tag{9}
$$

in which the first inequality is because $M_{i+1}$ is the maximal, and second inequality is due to the definition of SmI. Now the expectation reads

$$E\left[f_{x_{i+1}}(S_i)\right] \geq \frac{1}{k}\left\{\lambda_{S_i,|S^* \setminus S_i|} + E\left[f(S^* \cup S_i)\right] - E\left[f(S_i)\right]\right\}$$

If $f$ is monotonic, we can further lower bound $f(S^* \cup S_i)$ by $f(S^*)$ and proceed to induction for performance bound, however in the non-monotonic case, this lower bound does not stands any more. In this step the random choice of the algorithm becomes crucial: with lemma lemma:proba, we can show that on average, $f(S^* \cup S_i)$ still has a variant lower bound.

The trick is to notice that with the random greedy algorithm, in each iteration, any element $y \in V \setminus S_i$ will be selected into $S_{i+1}$ with probability at most $1/k$, hence at iteration $i$, $y$ stays outside of $S_i$ with probability at least $(1 - 1/k)^i$, or in other words,

$$\mathbb{P}\{y \in S_i\} \leq 1 - (1 - 1/k)^i = p$$

Define function $g(S) = f(S \cup S^*)$, then it is easy to see that $g$ is approximately submodular with $\lambda_{U,n}(g) = \lambda_{U \cup S^*,n}(f)$. Now let's apply the lemma to get

$$
\begin{aligned}
E\left[f(S^* \cup S_i)\right] = E\left[g(S_i \setminus S^*)\right] &\geq \left(1 - \frac{1}{k}\right)^i g(\emptyset) + \beta_{S_i \setminus S^* \cup S^*} \\
&\geq \left(1 - \frac{1}{k}\right)^i f(S^*) + \beta_{S_g}
\end{aligned}
$$

The last inequality is because $S_i \setminus S^* \cup S^* = S_i \subseteq S_g$, and $\beta_S$ is decreasing in $S$ (as a linear combination of $\lambda_{S,2}$). Continuing with this lower bound on $E\left[f(S^* \cup S_i)\right]$, we get

$$E\left[f_{x_{i+1}}(S_i)\right]$$
$$\geq \frac{1}{k}\left\{\lambda_{S_g,k} + \beta_{S_g} + \left(1 - \frac{1}{k}\right)^i f(S^*) - E\left[f(S_i)\right]\right\}$$

Define $\lambda_{S_g,k} + \beta_{S_g} = -\xi_{S_g}$ a constant with given $k$, then rearranging yields

$$E\left[f(S_{i+1})\right] - E\left[f(S_i)\right]$$
$$\geq \frac{1}{k}\left\{\left(1 - \frac{1}{k}\right)^i f(S^*) - E\left[f(S_i)\right] - \xi_{S_g}\right\} \tag{10}$$

$$E\left[f(S_{i+1})\right]$$
$$\geq \left(1 - \frac{1}{k}\right) E\left[f(S_i)\right] + \frac{1}{k}\left(1 - \frac{1}{k}\right)^i f(S^*) - \frac{\xi_{S_g}}{k} \tag{11}$$

The last inequality implies that the expected increments made by random greedy algorithm has guarantees, but is deteriorated by the lack of strong submodularity, whose negative effect is incorporated by $\xi_{S_g}$. Next, we will make use of this inequality with a induction framework and show the overall performance guarantee of the algorithm. Specifically, assume

$$E\left[f(S_i)\right] \geq \frac{i}{k}\left(1 - \frac{1}{k}\right)^{i-1} f(S^*) - \frac{\xi_{S_g}}{k}\sum_{j=0}^{i-1}\left(1 - \frac{1}{k}\right)^j \tag{12}$$

when $i = 1$, we have

$$kE\left[f(S_1)\right] \geq \sum_{x \in S^*} E\left[f(x)\right] \geq E\left[f(S^*)\right] + \lambda_{\emptyset,k}$$
$$\geq E\left[f(S^*)\right] + \lambda_{S_g,k} \geq E\left[f(S^*)\right] - \xi_{S_g}$$

where the first inequality follows because the first step choice $S_1$ is always maximum, the second and third inequalities are from the SMD definition and its decreasing property, and the last inequality is due to our worst case assumption that $f$ is not submodular and $\beta_{S_g} \leq 0$. Now assume (12) is true for any $i' = 1, 2, ...i$, then at $i + 1$ step, plugging into (11) gives

$$E\left[f(S_{i+1})\right]$$
$$\geq \frac{i}{k}\left(1 - \frac{1}{k}\right)^i f(S^*) + \frac{1}{k}\left(1 - \frac{1}{k}\right)^i f(S^*) - \frac{\xi_{S_g}}{k}\sum_{j=0}^i\left(1 - \frac{1}{k}\right)^j$$
$$= \frac{i+1}{k}\left(1 - \frac{1}{k}\right)^i f(S^*) - \frac{\xi_{S_g}}{k}\sum_{j=0}^i\left(1 - \frac{1}{k}\right)^j$$

which completes the induction. Let $i = k - 1$, we get

$$E\left[f(S_g)\right] \geq \left(1 - \frac{1}{k}\right)^{k-1} f(S^*) - \xi_{S_g}\left(1 - \left(1 - \frac{1}{k}\right)^k\right)$$

$$\geq \frac{1}{e} f(S^*) - \xi_{S_g} \geq \left(\frac{1}{e} - \frac{\xi_{S_g}}{E\left[f(S_g)\right]}\right) f(S^*)$$

∎

**Proof** Corollary 1

This is an easier case, we can start from last line of (9) and get

$$E\left[f_{x_{i+1}}(S_i)\right] \geq \frac{1}{k}\left\{\lambda_{S_i,|S^*\setminus S_i|} + E\left[f(S^* \cup S_i)\right] - E\left[f(S_i)\right]\right\}$$

$$\geq \frac{1}{k}\left\{\lambda_{S_i,|S^*\setminus S_i|} + E\left[f(S^*)\right] - E\left[f(S_i)\right]\right\}$$

since $f$ is monotonic, $f(S^* \cup S_i) \geq f(S^*)$. Rearranging yields

$$E\left[f(S_{i+1})\right] \geq \left(1 - \frac{1}{k}\right) E\left[f(S_i)\right] + \frac{1}{k} f(S^*) + \frac{\lambda_{S_i,|S^*\setminus S_i|}}{k}$$

$$\geq \left(1 - \frac{1}{k}\right) E\left[f(S_i)\right] + \frac{1}{k} f(S^*) + \frac{\lambda_{S_g,k}}{k} \tag{13}$$

Let's again use induction technique for clarity. Assume

$$E\left[f(S_i)\right] \geq \left[1 - \left(1 - \frac{1}{k}\right)^i\right] f(S^*) + \frac{\lambda_{S_g,k}}{k} \sum_{j=0}^{i-1} \left(1 - \frac{1}{k}\right)^j$$

Then one can easily check that this assumption stands for $i = 1$ with the definition and monotonicity of $\lambda_{U,m}$, and from $i$ to $i+1$ one can just plug the assumption into (13). Hence we have

$$E\left[f(S_g)\right] \geq \left[1 - \left(1 - \frac{1}{k}\right)^k\right] f(S^*) + \lambda_{S_g,k}\left[1 - \left(1 - \frac{1}{k}\right)^k\right]$$

Now if the function is submodular, we have $\lambda_{S_g,k} \geq 0$, then

$$E\left[f(S_g)\right] \geq \left(1 - \frac{1}{e}\right) f(S^*) + \left(1 - \frac{1}{e}\right) \lambda_{S_g,k}$$

$$\geq \left[1 - \frac{1}{e} + \left(1 - \frac{1}{e}\right)^2 \frac{\lambda_{S_g,k}}{E\left[f(S_g)\right]}\right] f(S^*)$$

where we have used $E\left[f(S_g)\right] \geq \left(1 - \frac{1}{e}\right) f(S^*)$ in the second inequality. On the other hand, if $\lambda_{S_g,k} \leq 0$, we get

$$E\left[f(S_g)\right] \geq \left(1 - \frac{1}{e}\right) f(S^*) + \lambda_{S_g,k}$$

$$\geq \left(1 - \frac{1}{e} + \frac{\lambda_{S_g,k}}{E\left[f(S_g)\right]}\right) f(S^*)$$

■

**Proof** Corollary 2
Simply note that in this case Lemma lemma:proba becomes $E\left[f(S(p))\right] \geq (1-p_1)f(\emptyset)$, and
we just follow the lines of proof of Theorem 3 with $\xi_{S_g}$ replaced by $\lambda_{S_g,k}$. ■

## 2. Causal Structure Learning

In this section, we connect the newly developed subset selection results to the problem
of causal structure learning for networks of processes: It is shown that, assume bounded
indegrees for each node, the structure learning problem can be reduced to solving OPT2 for
every process in the network. Hence the near optimal random greedy heuristic is applied
to establish an efficient algorithm for structure learning. Furthermore, we discuss directed
information estimation from streaming data, and propose a decomposition technique to
compute $\mathcal{I}(S^n \to \bar{S}^n)$.

### 2.1 Causal Structure Learning and its relation with Causal Subset Selection

A rich body of research exists in literature on the structure learning of graphical models for
i.i.d samples, however the problem becomes much more involved when we deal with non-
i.i.d dynamic networks of processes. Previously, the structure learning of dynamic networks
is usually addressed with multivariate regressive models. For example, in Materassi and
Innocenti (2010), the author proposed an algorithm to identify topology of network of
linear systems. In Bolstad et al. (2011), an alternative is proposed based on Group Lasso.
In this work, we adopt the result of a recently work Quinn et al. (2015), which defined the
notion of directed information graph, and proved its equivalence to minimum generative
models. First of all, the definition directed information graph is stated as follows,

**Definition 1** *Quinn et al. (2015) A Causal Graph with Directed Information as causality
metric, is a directed graph on V with each nodes representing a process, and there is a
directed edge from node $i \in V$ to $j \in V$, if and only if*

$$\mathcal{I}(X_i \to X_j || V \setminus \{X_i, X_j\}) > 0 \tag{14}$$

Compared to causal graph based on linear models, directed information graph is advanta-
geous in that (1) non-linear causality can be captured and Gaussian assumption is not re-
quired; (2) the graphical model is equivalent to generative models such as dynamic Bayesian
network Quinn et al. (2015); (3) confounders can be naturally eliminated due to the causally
conditioning in (14).

From the above definition, a naïve way of structure learning from data is to check
$\mathcal{I}(X_i \to X_j || V \setminus \{X_i, X_j\}) > 0$ for every pairs of processes in the network. This $O(|V|^2)$
algorithm seems viable in terms of computational cost, however, to estimate the causally
conditioned directed information, i.e., $\mathcal{I}(X_i \to X_j || V \setminus \{X_i, X_j\})$, the joint distribution
of all the processes in the network has to be estimated at first place. This requirement

---

**Algorithm 1** Structure Learning

$\mathbb{G} \leftarrow zeros(N, N)$
**for** $i \in V$ **do**
    $(a, \pi_i) \leftarrow \max_{j \in V} \mathcal{I}(X_j^n \rightarrow X_i^n)$
    $d \leftarrow a, m \leftarrow 1$
    **while** $d \geq \varepsilon$ & $m \leq k$ **do**
        $(a', j^*) \leftarrow \max_{j \in V} \mathcal{I}(X_{\pi_i \cup j}^n \rightarrow X_i^n)$
        $d \leftarrow a' - a, a \leftarrow a', \pi_i \leftarrow \pi_i \cup j^*$
        $\mathbb{G}(j^*, i) = 1, m \leftarrow m + 1$
    **end while**
**end for**

---

produces serious problems because high dimensional joint distribution is usually hard, if not impossible, to estimate without extra assumptions Negahban et al. (2009).

The remedy is to realize the following property

**Lemma 6** *In a directed information causal graph* $\mathbb{G} = (V, \mathcal{E})$*, let* $\pi(i) \in V$ *be the set of all parents of node* $i \in V$*, then for any other set* $W \in V$*, we have*

$$\mathcal{I}(X_{\pi i} \rightarrow X_i) \geq \mathcal{I}(X_W \rightarrow X_i) \tag{15}$$

which essentially indicates that the complete parents set always has maximal causal influence on its child node (process). Thus, the structure learning problem can be reduces to solving

$$\underset{S \subseteq V, |S| \leq k}{\operatorname{argmax}} \mathcal{I}(S^n \rightarrow X_i^n) \tag{16}$$

for each node $i \in V$, assuming maximal indegree is $k$ for all nodes. According to Corollary 1, a near optimal approximate solution can be obtained with either random or deterministic greedy search. A deterministic version is summarized in Algorithm 1. Compared to pairwise edge detection, this algorithm only requires estimating joint distribution of dimension at most $k+1$, which is significantly smaller than $|V|$, the dimension of the full joint distribution.

## 2.2 OPT1 Decomposition and DI estimation

Let us take another look at OPT1, which involves solving the problem $\operatorname{argmax}_{S \subseteq V, |S| \leq k} \mathcal{I}(S^n \rightarrow \bar{S}^n)$. Although we showed that the objective is submodular and near optimal solution can be obtained with greedy algorithm, it turns out we still need to estimate directed information from a subset $S \in V$ to its compliments. Again, direct estimation requires joint distribution of all processes in $V$, which is problematic when $|V|$ is large. Here the remedy is to realize that directed information graph $\mathbb{G}$ actually provides a sparse representation of the joint distribution. With some algebra, we can find the following decomposition

**Lemma 7** *OPT 1 Decomposition*

$$
\begin{aligned}
\mathcal{I}(S^n \to \bar{S}^n) = {} & \mathcal{I}\left(\mathcal{C}_{\bar{S}}(S^{n-1}) \to \mathcal{C}_S(\bar{S}^n)\right) \\
& + \sum_t I\left(\mathcal{C}_{\bar{S}}(S_t); \mathcal{C}_S(\bar{S}_t) \mid \mathcal{C}_{\bar{S}}(S^{t-1}), \mathcal{C}_S(\bar{S}^{t-1})\right)
\end{aligned}
$$

where $\mathcal{C}_A(B) \triangleq \{X_i \mid X_i \in B, \exists X_j \in A, \mathbb{G}(i,j) = 1\}$ denotes the set of adjacent nodes from $A$ to $B$. Hence by utilizing the learned structure, the directed information estimation in OPT1 is reduced to the estimation of local jointly probabilities, which often times have much smaller dimensionality.

For directed information estimation, in this work we use an estimator recently proposed in Jiao et al. (2013), in as much as its fast convergence and mild assumptions on the process. Interested readers are referred to Quinn et al. (2011)Quinn et al. (2015) and the reference therein for other possibilities. The procedure consists of (1) estimate a universal probability assignment, say $Q$, for the processes under consideration. This is done through the well-known context tree weighting (CTW) algorithm. (2) estimate directed information from process $X$ to $Y$ with

$$
\hat{\mathcal{I}}(X^n \to Y^n) \triangleq \hat{H}(Y^n) - \hat{H}(Y^n \| X^n) \tag{17}
$$

where the causal entropy is estimated with

$$
\begin{aligned}
\hat{H}(Y^n \| X^n) &\triangleq -\frac{1}{n} \log Q(Y^n \| X^n) \\
Q(Y^n \| X^n) &= \prod_{t=1}^{n} Q(Y_t \mid X^t, Y^{t-1}) \\
\hat{H}(Y^n) &\triangleq \hat{H}(Y^n \| \emptyset)
\end{aligned} \tag{18}
$$

Under some technical conditions, it can be shown Jiao et al. (2013) that the above method converges to the true DI with $O(n^{-1/2} \log n)$ sample complexity, when $L_1$ norm is used as the distance metric.

## 3. More Results

As a more interesting case study, we applied the proposed structure learning method to the PM data set, which contains hourly record of fine particulate matter (PM2.5) for 36 measured locations in north California. The geographic distribution of these locations is shown in the left subfigure of Figure 1. And the constructed causal graph is shown in the right subfigure. In this context, the subset selection problem corresponds to selecting "pollution sources". We solve OPT1 using greedy algorithm, together with the directed information decomposition technique. Interestingly, we find out that the detected pollution sources are mainly commercial, industrial or transportation centers, such as node 25 (San Francisco) and 7 (Richmond in east bay). Moreover, most of the constructed causal edges are consistent with climatic and geographical implications, such as the edge $29 \to 24$ in the Monetary bay valley. These results show that the proposed causal structure learning method constitutes a promising tool for data driven sensor placement and source detection.

(a) 36 measured locations in north California

(b) Constructed causal graph and selected pollution sources (in red)

Figure 1: Case study: North California air pollution