[Reviews · NeurIPS 2016]

Reviewer 1

Summary

This paper discusses the problem of causal subset selection using directed information. The problem becomes one of maximizing a submodular (or approximately submodular) function of choosing which features to use for prediction. The most interesting aspect is a novel definition of approximate submodularity and the obtained results on approximation that it yields.

Qualitative Assessment

There are really two issues with this paper. The first one is the discussion on how to use directed information for causality. Unfortunately, the word causality means two different things and these should be separated (and this discussion should be mentioned in this manuscript I think). The first concept of causality is really *prediction* of a time series using other time series. This prediction respects time and hence people call that causal. Given a bunch of time series we can ask which one can be used to better predict another (in the next time-step) and use this as a measure of causality. Then we can talk about finding the k-best predictors and get into 'causal' feature selection etc, as the paper does. The connection of this Granger-type of causality with directed information is interesting and was recently explored, e.g. in [20] as cited in this paper. This paper builds on this type of causality that perhaps should be called generalized Granger causality, or prediction causality or something like that. The second type of causality (that is in my opinion, and in the opinion of most people working in this area the correct one, as far as I understand) relates to counter-factuals (or the related structural equations). In this world, the key issue is what WOULD happen if I changed X, would Y change? This relates to understanding the functions or mechanisms that generate the data, rather than predicting. It is important to state in this paper that the best time-series prediction tells us nothing about how modifying one time-series will actually change the other. In short, I think this paper has quite interesting results but on prediction of processes and not really about causality. More specifically, the most interesting results are on submodularity and perhaps the authors should modify the manuscript to emphasize on the approximation of near-submodular functions (with Granger causality as an application). For the non-Granger causality literature, the authors should be perhaps mentioning the work by e.g. Pearl, Imbens and Rubin and Richardson. Theorem 2 is interesting and the proof seems correct. Lemma 1 is the key in showing how submodularity is relaxed in this paper (even though its called lemma 2 in the appendix ) and the proof is clear. Theorem 3 and its corollaries are the most interesting results in the paper I think, and they are really about a new relaxation of submodularity that is independent of causality (or any information metrics really) so that should be made clear in the paper. The second issue I'd like to raise is the following: Das and Kempe in [3] define another relaxation of submodularity called the submodularity ratio. It is incorrectly stated in this paper that [3] defined the concept of submodularity ratio for R^2 score only. In fact, the submodularity ratio is actually defined in [3],Def 2.3 for an arbitrary set function and then specialized for the special case of R^2. Furthermore, the performance guarantee that generalizes the 1-1/e result (Theorem 3.2 in [3]) actually holds for any set function with bounded submodularity ratio. This is stated in [3] before theorem 3.2 but admittedly the paper does not emphasize this general result enough in the abstract/intro. Now my main question is if (some of) the obtained results in this paper are also direct corollaries of Theorem 3.2 of [3]: This is because (if I understand correctly) the Submodularity index defined in equation (6) of this paper is the difference of adding elements one at a time vs adding them all at once. The submodularity ratio of [3] on the other hand seems to be exactly the ratio of the same quantities. Submodularity means the difference is greater than zero (lemma 1 of this paper ) which is equivalent to ratio >=1 (as stated in [3]). Submodularty ratio: sum fx(A) / f_S ( A) >=1 SMI defined in this paper sum fx(A) - f_S ( A) >=0 Also the relaxation of [3] is Submodularity ratio >= constant gamma and in this paper SMI difference negative but close to zero. Since the obtained results seem to be normalizing SMI/ f(S_greedy) it is possible there is a direct mapping from the results of [3] to this paper for general set functions. This is a non-trivial analysis for the authors to perform, so I would simply recommend that the authors discuss a possible connection to the ratio of [3]. The analysis is even more complicated since [3] only yields results for monotone functions while this paper obtains results for both monotone and non-monotone, so it would seem that the authors could be generalizing [3] beyond monotonicity which is very interesting. Minor Comments: [3] the author order should be reversed. Proofs of Proposition 2 and 3 where not clear (or i did not find them). The paper has many typos/ grammatical errors and there are some missing ?? in the Appendix. Examples of typos: Consider two random process X^n (processes) directed information maximization-> maximization problems (page 3) address in details-> in detail. any polynomial algorithm-> polynomial-time algorithm Throughout the paper: 'Monotonic' submodular functions are called *Monotone* we mainly analysis-> analyze we can justify its submodularity, (confusing sentence, please rephrase) that in literature-> in the literature (page 4) several effort-> efforts Overall I recommend that this paper is accepted after these issues have been addressed since the combinatorial optimization results are quite interesting.

Confidence in this Review

3-Expert (read the paper in detail, know the area, quite certain of my opinion)


Reviewer 2

Summary

The authors study two related problems which they refer to as source detection (OPT1) and causal covariate selection (OPT2). OPT1 is related to a body of work on sensor selection, but whereas past work typically use mutual information this work uses directed information. Likewise problems like OPT2 have been studied for decades, though often with strong modeling assumptions. The authors suggest using directed information for that too (it is a non-parametric method). The authors show that OPT1 is submodular but not monotone (thus greedy search has performance guarantees), while OPT2 is monotone but generally not submodular. The authors introduce a novel measure of how "submodular" a functions is (which they refer to as submodularity index SmI) and obtain performance bounds for maximization in OPT1 and OPT2 with cardinality constratints, where the bounds are a function of SmI. They also analyze both simulated and real world data, showing performance of the algorithm and computing some empirically observed SmI values.

Qualitative Assessment

Overall, I find the work promising and it contains a number of results which will appeal to the NIPS community. However, I think a significant revision for OPT2 is needed for it to appear at NIPS. They are also missing citations for work closely related to their work on OPT2. There are multiple minor issues that should also be addressed, listed further below. Major comments (What I like): -- Both problems are interesting and relevant to NIPS. Especially OPT2, broadly speaking, has been of great concern and generated decades of research in various domains. To approach it with a non-parametric method is quite appealing because then one needs not worry as much about strong modeling assumptions. -- SmI is an intriguing way of assessing submodularity, though it makes sense. It is highly context dependent (function of A and k), but that seems necessary given your later work and data analysis. I like how the results are not limited to specific functions but could potentially be applied to a broad range of combinatorial problems. -- The performance bounds are good. -- It was applied to both simulated and real world data. Aside from empirical values for SmI, it does give credence at least to employing greedy methods (especially when exact is computationally prohibitive). Major comments (What I think needs to be modified to improve the work): (A) My major concern is with regards to the usefulness of the bounds for OPT2. While results for OPT1 are clearly valid (since is submodular), the usefulness of the bounds for OPT2 is unknown. They are useful only when OPT2 exhibits "quasi-submodularity," but when does that happen? Statements such as (line 44) "is “nearly” submodular in particular cases" (line 216) "For quasi-submodular functions having negative SmI, although a degraded guarantee is produced, the bound is only slightly deteriorated when SmI is close to zero." (line 227) "Similarly, for quasi-submodular functions, the guarantee is degraded but not too much if SmD is close to 0. Note that the objective function of OPT2 fits into this category." are made throughout the paper but not justified or made precise. "quasi-submodular functions" is defined as (line 163) "We call a function quasi-submodular if it has a negative but close to zero SmI." The authors never give a precise definition of what consistutes "close." They also never give examples of when OPT2 is in fact quasi-submodular except in the extreme case of Proposition 2 (line 131) where you have independence *conditioned on Y_t*. [Please give an example that satisfies Prop 2. Even when causal covariates are marginally independent (which is already restrictive), say Y_{t+1} is a linear combination of marginally independent X_{it}'s, then conditional independence (prop 2) won't hold.] This is not to say that the formulas obtained are not useful for monotonic non-submodular functions (especially since they are applicable to potentially a wide range of combinatorial optimization problems), but some non-trivially "quasi-submodular" optimization problems need to be identified, and I don't think OPT2 is a good candidate. Perhaps for some certain classes of distributions it could be shown to be (and if the authors can do this, I would be happy with them keeping OPT2 in), but in general not. The graphical model literature on structure learning has examples where there is pairwise independence (similar to I(X-->Y)=0, I(Z-->Y)=0)) but the joint relationship (so I(X,Z-->Y) can be made arbitrarily large when the alphabet is not restricted (eg using the XOR function and converting vector valued binary variables (with Y(i) = X(i) XOR Z(i), then convert the vector to a larger-alphabet variable)). Also, note that while I am happy to see they applied their work to data, and that the randomized greedy did well, the fact that randomized greedy did well does not imply quasi-submodularity holds. [Greedy algorithms (though not randomized greedy) have been widely used in graphical model structure learning literature, for random variables and processes.] The experiments do provide evidence for using randomized greedy, which is good. I was furthermore happy that they included some empirical values of SmI index. And for row 2 of Table 1, since all the values are larger than -0.6321 the bound in corollary 1 (line 224-225) would be nontrivial (approximation ratio bigger than 0). This gives a little evidence of quasi-submodularity, and it is interesting that k=2 is the worst, whereas others have positive or "close" to 0 values. In addressing this major concern, if proving when quasi-submodularity holds (maybe you can define "close" to be relative to E[f(S^g)], such that close means bound in 224-225 is nontrivial) is too difficult, empirical evidence is fine, though it would need to be more extensive than for a single stock in one dataset. (B) The paper is missing a reference that appears to be closely related, especially for OPT2. Searching "Submodularity directed information" on arxiv yielded two works * [highly relevant] "Bounded Degree Approximations of Stochastic Networks" http://arxiv.org/abs/1506.04767 by authors Quinn et al (you cite some of their other works) which appears related for the OPT2 problem. Especially Section 5 of the arxiv version, as they do cardinality constrained causal covariate selection (OPT2; it looks like they are doing "approximation", though the objective functions look similar) and propose a approximate submodularity measure and obtain bounds on the deterministic greedy search. Their approximate submodularity measure is not the same as in this submission (SmI) and I can't tell whether the one in Quinn et al would be useful for studying OPT1. Also, wrt my comment above about "quasi-submodularity" being unjustified, this work by Quinn et al also does not theoretically justify when their approximate submodularity is useful (when their alpha is close to 1). They mention the arxiv version (I don't see it published yet) is based on a conference paper "Optimal bounded-degree approximations of joint distributions of networks of stochastic processes" ISIT 2013, which looks to have the same results (regarding approximate submodularity), though the causal covariate selection looks a little different. * [less relevant] a very recent paper (just a few months out before the NIPS 2016 initial deadline) "Submodular Variational Inference for Network Reconstruction" http://arxiv.org/abs/1603.08616 that reconstructs the network, but does so assuming a diffusion model. Minor comments -- mentioned in Major comment (A), please provide examples when Prop 2 holds, esp if you cannot more generally identify when OPT2 is quasi-submodular. -- Question: This work employs the CTW algorithm. But the available code http://web.stanford.edu/~tsachy/DIcode/download.htm and the original paper [9] are for pairs of processes. In the submission, you are dealing with sets. Did you modify the code to handle DI for sets? If so, are you sure the theoretical guarantees extend? Or did you do a mapping, such as converting three binary processes into one process of alphabet 8? This might be straightforward but I wonder if there would be performance hit (for finite length data). I'm assuming you did not just find pairwise relations for the original processes and then combine individual parents, as that could lead to significant errors. -- Can you provide any guidance to the reader on selecting $k$? I think this is important given that not only there are computation tradeoffs but the approximation coefficient is sensitive to $k$ (especially for OPT2) -- Line 185 with Algorithm 1. You should provide a sentance or two explaining how this random greedy relates to deterministic. It is not obvious at first glance of the notation, even though the idea is simple. "At each step, instead of picking the best, we pick one of the top k best" or something like that. -- line 131 "any" can have multiple meanings. I assume you want it to mean "every pair" though it can also mean "at least one pair". I'd suggest using "every" to avoid confusion. -- Provide references for quantities in Section 2. At first I thought you were introducing them but they are in some works you cite. -- Double check your Reference section; right now it is a little sloppy (eg line 313 "gaussian" should be capitalized, journals in 340 and 346 should be capitalized, conference title in 358 should be capitalized, you use anacronyms for some conference proceedings but then write out names for others) -- Lemma 2 line 197 please clarify the notation. Are these lower case y's processes? If so why lower case? And is Y={y_1, .. y_M} or a strict subset or a strict superset? -- in line 201 you make a comment about being able to lower bound SmI for order 2, yet wasn't that the worst case you found empirically? (thus suggesting it would be challenging to do so) -- fix grammer line 205 "with each element appears in" -- formula below line 201, what is "p_1" -- line 238, in formula for \gamma, if S^{g_k} means |S^{g_k}|=k, then just use k in the formula, and you might consider putting a table or graph to show behavior -- line 249 the data you use is old -- if it appeared in another paper you should cite the source. -- Section 5 for the data, I'm sure the stock data is not binary, perhaps you designed the Bayesian network data to be so; but describe how your discretize the data (especially since to just binary, not even quantizing) -- line 257 you should not say "is sufficient" for real world data unless you have ground truth knowledge you are basing that off of. Even convergence is not sufficient (maybe for length 5 it converges to one value, for length 6 another) -- line 261 include an equation reference; "the 1/e bound" is vague

Confidence in this Review

3-Expert (read the paper in detail, know the area, quite certain of my opinion)


Reviewer 3

Summary

The basic idea of the paper is to formulate the two problems, source detection and causal covariate selection, into cardinality constrained directed information (DI) maximization problems. It is then shown that the objective function of source detection which is precisely the DI from selected set to its complement is submodular, although not monotonic. It is also shown that the problem of causal covariates selection which is the DI from selected set to some target process is not submodular, however, the authors define a new notion called "near-submodularity" which holds in this case (at least in some problem instances). A novel quantity, namely submodularity index (SmI), for general set functions is introduced to measure how "submodular" a set function is. for a non-submodular functions SmI is negative. But if SmI is not so small, then the algorithm random-greedy will still perform near-optimally (even for non-monotone functions). As a separate contribution, the authors derive near-optimality bounds of the performance of the random-greedy algorithm based on SmI. Such bounds are then incorporated into analysing the behaviour of random greedy on the two main problems considered in the paper (i.e. source detection and causal covariate selection). In general, computing SmI may be intractable (exponential in size of the ground set). Hence, the authors provide a simple lower bound for SmI for the two problems considered. These theoretical results are then verified by numerical experiments.

Qualitative Assessment

The paper is clearly written in most parts and I enjoyed reading it. Apart from proposing efficient algorithms for the two problems considered (source detection and causal covariate selection), I believe that a separate contribution of the paper is to show the near-optimality of the random-greedy algorithm in terms of the so-called submodularity index (SmI). As a result, for functions that are "approximately" submodular (i.e. have a small SmI), the greedy algorithm may work well. Computing the value of SmI is exponential in terms of the size of the ground set. I suggest that the authors: (i) provide computable bounds on this index (as done in lemma 2) for other classes of se functions, (ii) provide some more applications where SmI is small. I should also mention that the bound provided in Lemma 2, becomes useless as n growrs as the mutual information terms will most probably grow by n. As a result, I suggest that the authors investigate the performance of random greedy on problems (lets say source detection) with the ground set size |V| being large (100 or so). It would be interesting to see how the proposed methodology compares to other algorithms in this setting. Also, in the experimental results, can the authors explain why they have not made any comparison with the other methods (e.g [21-22])? some minor comments: Line 60: "i" should be "t" or vice versa. Line 101, you never formally define what S^n is. Line 161: perhaps strictly-submodular? Line 280: should be "reduced".

Confidence in this Review

2-Confident (read it all; understood it all reasonably well)


Reviewer 4

Summary

The authors attempt at two submodular optimization problems motivated by directed information maximization. The directed information is used as a measure of causality. Although this is in general not accepted as a correct causality measure, especially when more than two variables are involved (such as in the example application of learning causal graphs at the end), it is still widely used and preserves its relevance in certain problems. Then authors move to consider a generic non-monotone submodular function, and define a measure of "submodularness" based on the the collective diminishing value of the derivative of the function, which they call submodularity index, SmI. This metric allows them to obtain potentially tighter bounds on the optimum value of the submodular maximization problems, although the bounds have a complicated relation with the result of the randomized greedy algorithm, and the gain is not immediately clear. The problem statements are not well justified, though the problems are theoretically interesting. The submodularity sections are interesting on their own without a causal interpretation. Authors are not careful in using the notion of "causality". Directed information is not a generally accepted notion of causality and in general it cannot be used to predict the effect of an action (intervention) on the system.

Qualitative Assessment

I would suggest authors to pose the paper as a submodular optimization paper, with applications in directed information maximization. This is because the main technical contribution of the paper is not in the causality literature, but in the submodular optimization. Also, this way, authors will avoid a discussion on whether directed information is the correct metric for causality or not. In the second paragraph of introduction, authors completely ignore a big part of causality literature, conditional independence-based, score-based learning methods, and methods based on assumptions on the structural equations. Please cite Pearl's or Glymour's books as a relevant reference, and justify why directed information is an alternative to these in dynamic systems. The problem formulation is not well justified and some claims are even incorrect: They state "The optimal locations can be obtained by maximizing the causality from selected location set S to its complement \bar{S}". 1) Please avoid using "causality" as a measure. Use "directed information". This is especially important since the causality measure of directed information, as used here, is not well accepted by all the researchers in the field. 2) By optimal, authors refer to the "source sites" of the pollution. This is even used in a case study only provided in the Supplementary material. This claim is incorrect. If the objective is to find the sources of the pollution, i.e., the set of nodes that cause the remaining set of nodes in a causal graph, using directed information will yield incorrect results. Consider three nodes that form a collider, basically X->Z<-Y. There is no reason to assume that choosing X and Y will maximize the directed information to Z. Hence this approach would not reveal the source nodes of the graph. I recommend authors to rephrase their objective. The second problem is to maximize the directed information from a set of nodes to another set of nodes, which is a generalization of first problem, which is termed as "causal covariate selection". However this formulation has the same problem described above. These are not the mere result of authors using a different notion of causality. In the first paragraph, authors give the example of advertisers choosing opinion leaders to maximize the influence of their ads. Using a node for advertising is a type of "intervention". And if the directed information does not yield the causal sources of the network, intervening on a child will not have any impact on its parents. Hence authors should be very careful not to make any claims about the result of a possible intervention, using directed information as their metric. From the motivation of the problem, we know that the set S contains variables that "influence" or "summarize" other variables. From this motivation, I expect the underlying causal graph to have subgraphs of the form s1->y<-s2. This is the type of causal relation that violates the condition in Proposition 2. Hence, given the setup of the problem, it is very unlikely that s1 and s2 will be independent given Y. (7) defines the submodularity index (SmI), which is the main novel metric in the paper for measuring the "submodularness" of a set function. The minimization problem to find SmI requires searching through two disjoint subsets S and A. Proposition (3) claims (no proof given) that the minimized term is a supermodular function. And concludes that SmI can be computed using tools for minimizing a supermodular function with cardinality constraints. However there is one problem: The definition of SmI requires a search over both S and A. Hence, solving for SmI does not directly become equivalent to a supermodular minimization but exponentially many supermodular minimization problems. I suggest authors explain why Algorithm 1 runs in time O(nk): Simply say calculating M_i does not require a search, since objective is a modular function of elements of M_i. In the simulations: Causal Subset Selection Results: I believe "the 1/e reference bound" refers to e\times Random Greedy whereas Upper bound by SI refers to Random Greedy/(1/e+Delta) where Delta is the additional term in Theorem 3. Please clearly state this in the paper. In the causal structure learning application: Authors do not define the causal graph in the main paper (only in the appendix). It is important to define what causal graph means in this context since there is no universally accepted notion of a causal graph. From the appendix, it is clear that this framework only works in the "directed information graphs". From a Pearlian perspective, it is expected that an approach based on directed information would not be sufficient for learning (even the parts of) the causal graph. Hence, please clearly state that the application is only for learning directed information graphs in the paper. In general, the theorem/lemma labelings in the paper does not match with the ones in the supplementary material: The labeling of Theorems 1 is not correct. All theorems lemmas are numbered together in the supplementary file starting from 1, whereas labeling in the paper has a consistent numbering. Please fix this for easier cross referencing. Notation used in the proof of Theorem 2 is slightly confusing and inconsistent. Authors use \bar{A \cup {y}}_t to refer to the set of variables in the complement of A union {y} at time t. This format might be confused with y_t. Inconsistently, they also use A^t\cup {y}^t instead of (A\cup{y})^t In the manuscript. I recommend using extra space between variable set and t, or using |_t to clarify the set. Also I(X,Y|Z) should be I(X;Y|Z). The proof of Theorem 2 is correct. Please use \coloneqq (or similar) to distinguish equalities from definitions. It will make paper easier to follow. The proof of Proposition 1 is correct, although the authors skipped a few of the last steps. The proof of proposition 2 is not clear. What is causal naive Bayesian? Please provide a complete proof. Please provide the proofs for Proposition 3 in the supplementary material. Lemma 1 of paper is labeled as Lemma 2 in the supplementary material. What is "deviance" mentioned in this proof? The proof is not complete in its current form. \ref{.} is missing from "lemma:proba" in the proof of Theorem 7 (Theorem 3) in the supplementary material. Similarly in the Proof of Corollary 2. Also there is ??'s in Section 2 of supplementary material. There are also some typos in the supplementary material (he->the), please proofread.

Confidence in this Review

3-Expert (read the paper in detail, know the area, quite certain of my opinion)


Reviewer 5

Summary

The paper addresses the problem of identifying subsets of causal sources by introducing a novel measure for quantifying the degree of submodularity. To apply greedy algorithms to this type of problem, the authors derive the submodularity index (SmI), a measure that allows them to provide bounds on a random greedy algorithm as a function of the SmI. Thus, this work generalises the well established approach of submodular function maximisation. Using these results, the authors study the problems of source detection and causal covariate selection (source/destination covariates where one is the complement of another) by the information-theoretic measure directed information (DI). The first problem is shown to be submodular but not monotonic, while the second is nearly submodular (according to the SmI). The theoretical results are evaluated using DBN's Murphy's Bayes net toolbox for generating datasets and revealing the underlying causal structure.

Qualitative Assessment

The paper covers several fundamental and relevant topics, and provides potentially interesting results for each topic. Unfortunately, this broad scope dilutes the impact of the paper; the contribution of the work as a whole is incoherent. The notion of the SmI introduced in Sec. 4, and in particular, the implications of Theorem 3 are of relevance to a broad class of NP-hard problems. However, the authors restrict their application of SmI to causality analysis. Although the field of causality analysis is significant in its own right, I feel there is insufficient analysis of the fundamental properties of SmI prior to presenting this particular application. The paper reads as if its main focus is on the causal subset selection component instead. The effect of the paper's broad scope is that it suffers in depth, and consequently is difficult to understand given only the material i the body of the paper. The first 5 pages present fundamental results on optimisation of (nearly) submodular set functions, but the proofs are not included in the body of the paper. These proofs are integral to the results of the paper and should occupy a more prominent position. It could be better to condense the analysis of the causal subset selection component to make room for this; I don't feel there is scope for both inside a single conference paper. Minor comments/questions: - You might also consider transfer entropy as a measure of predictability, for which I imagine you will have similar results for the SmI. DI was originally intended to be attributed to feedback structure (according to Massey). [2] is a good reference for TE over DI in certain situations. - Need references for directed information and causally conditioned entropy in Sec. 2 - Line 6 do you substantiate the idea or quantify it? - Line 8 "the random" or "a random" or "we propose a random" - Line 9 "guarantee" -> "guarantees" - Line 50 appears to be the first mention of SmI in text. It is not expanded prior to this but is expanded later (line 135-6) - Line 70 "a a" - Line 140 "several effort has" -> "several efforts have" - Line 143 "existing works" -> "existing work" - Line 151 "indexes" -> "indices" - Line 120 "detour" should probably be "digression" - Line 229 should SmD be SmI? - Perhaps a definition of monotonicity is required (at least in Supp. Mat.) - The references appear to be below the allowed font size of \small - References inconsistently use acronyms for some conferences (and abbreviations for some journals) and full titles in others - Line 286 Clarify under which conditions this maximum DI finds the underlying graph structure. I assume for fully observed DBN's? - Line 286 "can be reduces" -> "can be reduced" - Line 305 "an novel" - Line 313 "gaussian" -> "Gaussian" - Line 331 "dc" -> "DC" - Line 340 "The Bayes"; Inconsistent use of "et al."; "matlab" -> "MATLAB - Line 350 Need periods after abbreviations, e.g., "Comput" -> "Comput." - (Supp. Mat.) Lemma 5 "definition (??)" is probably Definition 2? References: [1] A. Krause, D. Golovin. Submodular Function Maximization. Tractability: Practical Approaches to Hard Problems 3.19 (2012): 8. [2] M. Wibral, R. Vicente, and J. T. Lizier. Directed information measures in neuroscience. Heidelberg: Springer, 2014.

Confidence in this Review

2-Confident (read it all; understood it all reasonably well)


Reviewer 6

Summary

This paper studies causal subset selection based on directed information. Two tasks are formulated and their sub-modularity properties are explored through a so called sub-modularity index introduced by the authors.

Qualitative Assessment

This paper investigates the subset selection problem with respect to directed information as the causality measure. The main contribution comes from introducing a sub-modularity index to indicate how close the function is to sub-modular. The index seems to be natural and it is a bit surprising that it is not been studied before. Overall, the methods are novel and interesting. The application of the method, learn the causal structure, is quite interesting.

Confidence in this Review

1-Less confident (might not have understood significant parts)